# Mental Health in Healthcare Workers Post-COVID-19: A Latin American Review and Insights into Personalized Management Strategies

**DOI:** 10.3390/jpm14070680

**Published:** 2024-06-25

**Authors:** Sandra Muñoz-Ortega, Daniela Santamaría-Guayaquil, Jade Pluas-Borja, Geovanny Alvarado-Villa, Verónica Sandoval, Rubén Alvarado, Ivan Cherrez-Ojeda, Marco Faytong-Haro

**Affiliations:** 1School of Health, Universidad Espíritu Santo-Ecuador, Samborondón 092301, Guayas, Ecuador; slmunoz@uees.edu.ec (S.M.-O.); dsantamaria@uees.edu.ec (D.S.-G.); jadepluas@uees.edu.ec (J.P.-B.); galvarado@uees.edu.ec (G.A.-V.); ccherrezo@uees.edu.ec (I.C.-O.); 2Research Center, Universidad Estatal de Milagro, Milagro 091050, Guayas, Ecuador; vsandovalt@unemi.edu.ec; 3Interdisciplinary Centre for Health Studies (CIESAL), Department of Public Health, School of Medicine, Faculty of Medicine, Universidad de Valparaíso, Valparaíso 2362735, Chile; ruben.alvarado@uv.cl; 4Research Department, Ecuadorian Development Research Lab, Daule 090656, Guayas, Ecuador; 5Research Department, Respiralab Research Group, Guayaquil 090512, Guayas, Ecuador

**Keywords:** mental health, healthcare worker, anxiety, depression, distress, Latin America

## Abstract

Over the COVID-19 pandemic, the impact of enduring mental health on healthcare workers has become increasingly evident. This review focuses on post-pandemic mental health challenges faced by healthcare personnel in Latin America. This highlights the persistent burden on healthcare workers, especially women, which is exacerbated by economic disparities, inadequacies in the healthcare system, and ongoing occupational stressors. Our literature review, utilizing databases such as PubMed, Scopus, and Google Scholar, scrutinized the mental health status of healthcare professionals in the region after the pandemic’s peak. The analysis indicated sustained levels of psychological distress, with frontline workers and women continuing to be affected disproportionately. These findings emphasize the urgent need for personalized interventions to effectively address the complex mental health challenges in this context. This review advocates strategic interventions, including tailored psychological support, innovative diagnostics, and technological solutions, integrated into patient-centered care models. Such approaches aim to enhance the mental resilience and overall well-being of healthcare professionals across Latin America in the post-COVID era.

## 1. Introduction

The coronavirus disease (COVID-19) pandemic has caused an unprecedented global crisis. Nearly four years after its onset, it is estimated that there have been more than 760 million cases and 6.9 million deaths reported worldwide [1]. Although the majority of the infected individuals survived, a range of health effects, particularly on mental health, continued to manifest. The pandemic left quarantined communities facing fear of an unknown disease and vulnerable to various psychological disorders. Changes in daily activities, emotions, and relationships were evident to all, with children and adolescents being particularly susceptible. However, it is clear that healthcare workers, playing a crucial role in the care and attention of sick people, are one of the population groups with the greatest impact on their mental health, with a high prevalence of anxiety, depression, and other mental disorders [2].

During the pandemic, healthcare workers faced a series of challenges that had not been experienced before. Concerns related to insufficient resources, lack of training, lack of organizational and work support, inaccurate and inconsistent information, and poor organizational preparedness are recurrent and extremely disturbing problems [3]. Additionally, social distancing measures have hindered healthcare workers’ access to emotional support from family, friends, and colleagues [3,4]. The most prevalent psychological disorders among healthcare personnel include burnout syndrome, chronic fatigue, depression, and anxiety [5,6]. In response to this unprecedented situation, tools have been developed to safeguard mental health, such as enforcing quarantine only for as long as necessary, providing clear rationales and detailed protocols for quarantine, and ensuring that adequate supplies are available. However, few strategies specifically address the unique needs of healthcare personnel [7].

For instance, a comprehensive systematic review and meta-analysis revealed that during the pandemic, approximately 40% of healthcare workers experienced anxiety, while approximately 37% suffered from depression, highlighting the significant impact of the crisis on this group [8]. These findings highlight the urgent need for effective evidence-based mental health interventions tailored to the unique challenges faced by healthcare workers.

Additionally, numerous studies worldwide have highlighted the complexity and urgency of addressing the mental health of healthcare workers. A cross-sectional longitudinal study conducted in a nursing home in France found that one-third of nursing staff (n = 127) met the screening criteria for at least 1 mental health disorder. The most prevalent were panic attacks, affecting 22.05% of the staff, followed by probable depression (16.54%), post-traumatic stress disorder (10.24%), generalized anxiety disorder (9.45%), and substance use disorders (3.94%) [9]. In contrast, a qualitative study conducted among healthcare personnel in Pakistan revealed feelings of fear, loss, and frustration. One respondent stated, “I isolated myself twice because of a light cough. Every time I touch an elderly person, I think ‘okay, this is the moment I get the virus’… after my colleague also got infected, I felt feverish for many days”, illustrating the anxiety and fear of contagion that many healthcare staff members were experiencing [10].

In Latin America, the situation during the pandemic has been marked by additional challenges compared with other regions of the world. Multifaceted factors ranging from socioeconomic problems to deficiencies in public health structures have contributed to this reality. First, the economic disparities present in the region have significantly contributed to the variability in people’s ability to cope with the pandemic. In addition, differences in the quality and accessibility of health care between urban and rural areas have influenced the capacity of health systems to cope with crises, generating greater vulnerability in some populations [11]. Years after the beginning of the pandemic, its effects have persisted in certain Latin American countries, showing a high prevalence of post-traumatic stress disorder, particularly among women [12].

Acknowledging that research has primarily focused on the mental health of healthcare workers during the COVID-19 pandemic, our present literature review seeks to explore the less-examined post-pandemic period. Existing studies provide limited insights into the ongoing mental health challenges that persist beyond the immediate crisis. Our study aimed to review the available literature to first identify the current mental health status of healthcare workers in Latin America and the persistent challenges following the peak of the pandemic. Additionally, we aimed to gather and analyze information on strategic interventions and personalized approaches to improve the mental health of healthcare professionals in the region. We also advocate tailored psychological support and integration of innovative diagnostics and technological solutions in patient-centered care models. This effort underlines the significance of our study in contributing to the development of a sustainable healthcare workforce in Latin America during the recovery and adaptation periods following the COVID-19 pandemic.

## 2. Methodology

A review of the available literature was conducted using PubMed, Scopus, and Google Scholar. The search terms were organized into three main categories: 1. individuals working in the healthcare sector at any level of care (terms used: provider, personnel, professional, and workers); 2. scope, services, and units providing primary healthcare assistance (terms used: frontline, emergency, intensive, internal, and primary level); 3. mental health status, psychological, and emotional well-being (terms used: disorders, mental, stress, burnout, depression, and anxiety); and 4. study location in Latin America (terms used: Latin America and Hispanic). For each block, terms indexed in the Medical Subject Headings (MeSH) and Health Sciences Descriptors (DeCS) were identified and combined with free-text terms.

The search was conducted from 15 to 31 January 2024, including all publications available in the last 3 years related to the mental health of healthcare workers. Table 1 lists the strategies and terms used for each database, as well as the results obtained.

Once the articles were identified and duplicates were removed, a process of (pre)selection, choice, and analysis was performed. Inclusion and exclusion criteria were considered for document selection, as shown in Table 2. After reviewing the articles, only those that included data on the mental health of healthcare personnel in Latin America from 2022 to 2024 were selected for further analysis. This process resulted in the inclusion of 10 articles while 17 were excluded.

## 3. Development

Table 3 outlines the characteristics of selected articles examining the psychological and emotional consequences of healthcare workers post-SARS-CoV-19 pandemic, focusing on studies from Latin American countries and Spain, given the cultural similarity, published between 2021 and 2023 in Spanish or English, excluding mixed-method studies and non-healthcare-related topics. In a study conducted among obstetricians in Peru, 51.5% of the surveyed personnel had depressive symptoms and 90.8% had stress, with the largest number being women [13].

A total of 2 years after the onset of the COVID-19 pandemic in Colombia, an ad hoc study observed that healthcare personnel under the age of 28 had a 1.02 times higher risk of experiencing depressive symptoms than individuals aged 29 to 40 (95% CI: 1.24–3.30). Conversely, those aged over 41 had a 0.16 times higher risk than those aged 29–40 years (95% CI: 0.65–2.05). Healthcare workers in the middle socioeconomic stratum showed 1.00 times higher risk (95% CI: 1.16–3.45) of depressive symptoms than those in the high socioeconomic stratum, who exhibited a 0.72 times higher risk, although this difference was not statistically significant (95% CI: 0.61–4.84). Additionally, doctors displayed a 1.75 times higher risk (95% CI: 1.11–6.77) of depressive symptoms compared to nursing staff [19].

Regarding the impact of the work environment on mental health, a study in Colombia found that healthcare professionals in outpatient care had a 1.82 times higher risk of developing depressive symptoms compared to their counterparts in other settings. Furthermore, those working longer hours daily during the COVID-19 pandemic faced a 1.80 times greater risk of depressive symptoms than those with shorter work hours. Additionally, tobacco users reported a 2.51 times higher likelihood of experiencing depressive symptoms than did non-users [19].

In Argentina, an exploratory and descriptive study was conducted through interviews with 53 workers from three types of rehabilitation services (focused on supporting socialization, work, or housing) at 2 time points (end of 2020 and end of 2021). The study revealed a broad and enduring impact on services that became less accessible and effective in contributing to rehabilitation. Common aspects were observed, such as adaptation to continued functioning, centrality of technology, and subjective impact on workers, characterized by fatigue. One year after the pandemic, two-thirds of socialization-oriented services were greatly affected or closed, and levels of stress and anxiety among healthcare personnel increased by 5% [14].

Another study conducted in Brazil in 2022, which involved a sample of 702 healthcare professionals, found that the prevalence of common mental disorders was 43.2%, higher among those with symptoms of previous mental disorders, and that there was an association with workload overload [20].

A descriptive exploratory study conducted among 131 psychologists working in hospitals in Brazil revealed a widespread impact on work among almost all participants. It highlighted the necessity for professional preparation to adapt to this new scenario and perceived lack of institutional support, and nearly half of the studied population reported experiencing symptoms of significant psychological distress since the onset of the pandemic [12,17].

A study conducted in Uruguay, targeting healthcare personnel, administered a test to explore the participants’ subjective experiences. The results revealed that the most frequently mentioned theme was working conditions, followed by work organization and personal experiences. Subcategories, such as anxiety and stress, fear, fatigue, lack of social support, and workload, were among the most frequently mentioned. This study emphasizes the importance of creating safe and healthy work environments at both organizational and health policy levels [16].

In Chile, evidence has shown that symptomatology is higher when there is direct involvement in the care of COVID-19 patients, with an increased risk of developing post-traumatic stress symptoms. This psychological distress can impact attention, comprehension, and decision-making abilities, implying a deleterious effect on overall well-being [23].

In Ecuador, a descriptive cross-sectional study involving 400 professionals from various healthcare fields was conducted using a sociodemographic form and the Hospital Anxiety and Depression Scale. Of the total professionals studied, 46.40% exhibited anxiety and 34.50% had depressive symptoms. The highest scores for anxiety and depression were reported by nursing personnel [13].

In Spain, data revealed a high percentage of healthcare professionals experiencing symptoms of anxiety, stress, depression, and sleep disorders. The prevalence of anxiety, depression, and stress was higher than that reported in previous studies on COVID-19 [19,22]. Severe levels of anxiety are even higher than those found in China, where the pandemic started [24]. A study in Spain also revealed higher levels of anxiety, depression, stress, and insomnia among female healthcare personnel, with statistically significant differences in anxiety and stress, which is consistent with previous studies [25]. The greater tendency towards internalizing symptoms in women, supported by previous research, may stem from their primary role as caregivers in households (for children and parents), which heightens their anxiety and stress due to the fear of contagion [22,25,26].

One of the first Spanish studies presented a significant worsening of self-perceived health status, quantity, and quality of sleep hours, a 30% rate of sick leave among nurses, and a 20% rate among doctors, with 57% of doctors reporting physical exhaustion and 48% reporting emotional exhaustion. Additionally, there has been an increase in the contemplation of leaving the profession and early retirement [18].

Another study conducted in Spain, through an anonymous online survey with occupational and non-occupational variables, Goldberg’s anxiety and depression questionnaire, and the SF-12 conducted in 2023, showed that 58.1% of the surveyed personnel presented anxiety compared to 46.5% in 2022. The prevalence of depression is 39.5% in 2021 and 37.2% in 2023. The increase in anxiety scores was statistically significant (*p* < 0.001). In 2021, the perceived quality of life was 42.4 ± 13.0 vs. 47.1 ± 11.8 in 2022, showing a significant improvement (*p* < 0.033) [27,28].

## 4. Discussion

In addition to the relative scarcity of published studies, there was an overall approach, not always comparative, in the analysis of data without discriminating between frontline professionals and other healthcare personnel. In this regard, this review highlights the need for a rigorous understanding of the sociodemographic and professional profile of the personnel directly involved in providing healthcare to COVID-19 patients. The prevalence of depressive symptoms has increased post-pandemic, particularly among physicians and women. Notably, younger age correlates with a higher predisposition to depressive symptoms, possibly due to less-developed coping skills in stressful situations [29].

In times of social and political crisis, it is common to attribute all emotional suffering to “mental disorders” or, worse yet, to “mental illness”, contributing to the medicalization and psychiatrization of the population. This trend is even observed among healthcare professionals, especially those working in emergency services, intensive care units, and primary care. For them, an increasing incidence of “mental disorders” is forecasted, although specific terms such as “professional burnout” or “burnout” burnout, may sometimes be used. It is undeniable that poorly managed emotional and psychosocial pressures are the root cause of burnout, although they may (or may not) be based on underlying mental disorders [30,31].

From the perspective of grief and trauma theory, it is crucial to recognize that the effects of a stressor or trauma vary not only according to its intensity, duration, and characteristics but also according to the psychosocial context in which it occurs. Moreover, these effects are influenced by gender and psychological characteristics of the individual or group affected by trauma or stress. At the primary care level, morbidity and mortality have a significant impact. Illnesses, absenteeism, shift changes, and excessive workloads directly affect the emotional state and mental health of professionals, with healthcare workers being the most affected [22].

As detailed in our review, the mental health challenges faced by healthcare workers in Latin America during the COVID-19 pandemic are not only severe but multifaceted, influenced by socioeconomic disparities, occupational stress, and systemic healthcare inadequacies. The prevalence of anxiety, depression, and stress among this demographic underscores the urgent need for targeted interventions beyond generic approaches. These findings catalyze the transition from identifying problems to implementing solutions. In the following section, we focus on actionable strategies and tailored interventions grounded in the principles of personalized medicine. These proposed strategies are designed not only to mitigate the current crisis, but also to fortify healthcare workers against future systemic shocks, ensuring their mental well-being and, by extension, the efficacy of healthcare systems across Latin America.

Building on a detailed review of the severe and multifaceted mental health challenges faced by healthcare workers in Latin America during the COVID-19 pandemic, it is evident that these issues are part of a global pattern. Although socioeconomic disparities, occupational stress, and systemic healthcare inadequacies in Latin America are pronounced, the prevalence of anxiety, depression, and stress among healthcare workers is a universal concern.

## 5. Global Comparison of Mental Health Impacts on Healthcare Workers

Mental health challenges faced by healthcare workers during and after the COVID-19 pandemic are a global concern. Although our study primarily focused on Latin America, contrasting these findings with data from other regions highlights the unique difficulties faced by healthcare workers in Latin America and emphasizes the global need for personalized mental health interventions. The following comparison underlines the urgency of strategies aimed at strengthening healthcare workers against future crises and enhancing the efficacy of healthcare systems worldwide.

### 5.1. Global Context

In Europe, the importance of whole-system changes in healthcare settings has been highlighted as crucial for enhancing staff health and wellbeing. A systematic review by Brand et al. emphasized the need for interventions that understand local staff needs, engage staff at all levels, and ensure strong and visible leadership [32]. This approach aligns with findings from Northern Italy, where a significant percentage of healthcare workers experienced moderate-to-severe PTSD and anxiety during the pandemic [33]. This review suggests that incorporating whole-system changes, particularly those involving leadership engagement and staff empowerment, could mitigate similar mental health challenges and improve the overall well-being of healthcare professionals.

### 5.2. Latin American Context

Our review showed that healthcare workers in Latin America face additional challenges due to socioeconomic disparities and systemic healthcare inadequacies. For instance, in Ecuador, the healthcare system encountered structural issues, such as fragmentation and segmentation, compounded by limited public resources and unfavorable economic conditions. These factors severely affected the response to the health, social, and economic crises triggered by COVID-19. The prevalence of anxiety, depression, and stress among these workers underscores the urgent need for targeted intervention. The financial response in Ecuador did not reflect the increased demands on the health sector, with budget cuts leading to reduced service provision, particularly for non-COVID diseases [34]. This situation starkly contrasts with the circumstances in European and Asian countries, where healthcare systems may be more robust, thus highlighting the amplified vulnerability of Latin American healthcare professionals.

### 5.3. Unique Challenges in Latin America

Mental health impacts in Latin America are intensified by factors such as economic instability and unequal access to healthcare. These issues contribute to higher levels of psychological distress among healthcare workers compared with their counterparts in more economically stable regions. In addition, limited resources and support systems exacerbate the challenges faced by these workers. Unlike regions with robust healthcare systems, Latin American healthcare workers often face shortages of essential medical supplies and personal protective equipment, leading to increased stress and anxiety. Furthermore, the lack of mental health services and stigma surrounding mental health issues prevent many from seeking the help they need, deepening the crisis. By contrast, healthcare workers in more developed regions often have access to comprehensive mental health programs and support networks, which can significantly mitigate the psychological burden of their work.

## 6. Interventions in Latin American Healthcare Personnel

Recent interventions aimed at supporting the mental health of healthcare workers in Latin America have demonstrated innovative approaches and promising outcomes. These initiatives, including online psychological interventions, telehealth services, and mindfulness-based practices, address the unique challenges faced by healthcare personnel during the COVID-19 pandemic. Each intervention leveraged technology and psychological techniques to mitigate stress, anxiety, depression, and burnout, enhancing the well-being and resilience of healthcare workers in this region.

### 6.1. E-Health Psychological Intervention for COVID-19 Healthcare Workers: Protocol for Its Implementation and Evaluation

The study “E-Health Psychological Intervention for COVID-19 Healthcare Workers: Protocol for its Implementation and Evaluation” describes a protocol for an online psychological intervention aimed at healthcare workers in Mexico impacted by COVID-19. This intervention, named “Personal COVID”, employs a randomized clinical trial with two modalities, self-administered and delivered by online therapists, covering nine core modules and three complementary ones. The objectives are to reduce anxiety, depressive symptoms, burnout, and compassion fatigue and to enhance quality of life and self-care. The estimated sample size was at least 49 participants, evaluated at four points: pre-test, post-test, and follow-ups at 3 and 6 months, using techniques from Cognitive Behavioral Therapy (CBT), Acceptance and Commitment Therapy (ACT), Mindfulness, and Positive Psychology to address a wide range of emotional and behavioral outcomes. Since this is a protocol, specific results of the intervention have not yet been reported; these are expected once the intervention is completed and the data are analyzed according to the established metrics [35].

### 6.2. Telehealth in Community Mental Health Centers during the COVID-19 Pandemic in Peru: A Qualitative Study with Key Stakeholders

The article “Telehealth in Community Mental Health Centers during the COVID-19 Pandemic in Peru: A Qualitative Study with Key Stakeholders” reveals the transition to telehealth in community mental health centers (CMHCs) in Lima and Callao, Peru, during the pandemic. Driven by government regulations and local adaptations, this transition included adopting technologies such as telephone calls, WhatsApp, and videoconferencing for consultations and follow-up. Despite the benefits, such as improved communication and continuous care access for users who moved to other cities, significant challenges were faced, including a lack of technological resources, communication saturation, and preference for in-person care. These findings underscore the need to strengthen technological infrastructure and consider human preferences in future mental health service planning to ensure effective and sensitive telehealth implementation [36].

### 6.3. The Impact of an Online Mindfulness-Based Practice Program on the Mental Health of Brazilian Nurses during the COVID-19 Pandemic

The study “The Impact of an Online Mindfulness-Based Practice Program on the Mental Health of Brazilian Nurses during the COVID-19 Pandemic” demonstrated that an online mindfulness program was effective in improving the mental health of Brazilian nurses. With a 70.12% adherence rate, the program significantly reduced perceived stress, anxiety, and depression among participants, decreasing stress levels from 29.5 to 22.5, anxiety from 35.0 to 28.0, and depression from 11.0 to 6.5%, respectively. Additionally, an increase in both unidimensional and multidimensional mindfulness was observed, reflecting an enhanced ability to concentrate on and be aware of the present. Participants expressed high satisfaction with the program, noting relief in physical and mental overload and an improvement in well-being, life satisfaction, and work, underscoring the utility of these interventions to support the mental health of professionals in challenging contexts [37].

## 7. Personalized Mental Health Management for Healthcare Workers

### 7.1. Personalized Interventions

Understanding the unique stressors and challenges faced by different roles in health care is essential for developing targeted support programs. For instance, nurses often face intense, high-stakes environments and could benefit from resilience training tailored to the scenarios they encounter frequently, such as managing patient crises or end-of-life care situations [38]. Doctors dealing with the burden of critical decision-making and long hours might require stress management workshops that focus on mindfulness and decision fatigue [39]. Administrative staff, often overlooked, can be supported through programs addressing workplace organization, communication efficiency, and conflict resolution [40].

Research has shown that personalized psychological interventions are not only feasible but also highly effective. For example, a systematic review highlighted by Nye, Delgadillo, and Barkham found that personalized interventions substantially improved psychological outcomes compared to generic programs [41]. Similarly, Hornstein et al. in their study noted that incorporating personalization strategies in digital mental health interventions specifically designed around patients’ symptoms and needs could lead to better management of depressive symptoms among healthcare workers [42].

Additionally, the integration of adaptive treatment models that tailor interventions to an individual’s changing needs and capacities, as discussed by Bennett and Shafran [43], can enhance the effectiveness of these interventions by ensuring that they remain relevant over time.

Personalized interventions in healthcare settings are vital. They must be thoughtfully designed to address the specific challenges faced by different groups within the healthcare workforce, as supported by evidence from research that underscores the benefits of such tailored approaches. This not only ensures the well-being of healthcare workers but also enhances their capacity to deliver optimal patient care.

### 7.2. Innovative Diagnostic Approaches

The integration of innovative diagnostic tools in healthcare settings can substantially enhance the early detection and management of mental health issues among healthcare workers. Advancing advanced technologies such as genetic testing and biomarker analysis can facilitate the development of personalized care plans tailored to individual stress responses and genetic profiles. For example, a study by Gu et al. [44] developed a genetic biomarker-based diagnostic model using machine learning techniques to predict major depressive disorders, showcasing the potential of genetic assessments in mental health diagnostics.

Furthermore, the use of machine learning models to analyze personal and occupational data can revolutionize preventative strategies and predict mental health risks before they become apparent. Adler et al. advocated the development of digital biomarkers for mental health, emphasizing the role of open data in enhancing predictive models [45]. This approach could lead to earlier interventions and more effective mental health management among healthcare professionals.

Additionally, the application of RNA editing-based biomarkers has been explored as a method to refine diagnoses in psychiatric conditions such as bipolar disorder, further illustrating the impact of innovative diagnostic approaches on mental health care [46].

Overall, the integration of these novel diagnostic tools promises to transform the landscape of mental health care within healthcare settings, providing timely and precise interventions tailored to the specific needs of healthcare workers.

### 7.3. Technological Solutions

Digital health technologies, including mobile health apps and telepsychiatry, present a transformative approach to providing scalable and personalized mental health support, which is especially beneficial in remote or underserved areas. These technologies enable healthcare workers to access vital mental health services that might otherwise be unavailable owing to geographic and resource limitations.

For example, mobile health apps facilitate the continuous monitoring and management of mental health by using adaptive algorithms that personalize interventions based on user feedback and progress. Customization can significantly improve engagement and treatment outcomes. Torous and Roberts [47] highlighted the potential of mobile health apps to transform mental health care by providing accessible, personalized, and scalable tools.

Telepsychiatry extends the reach of mental health professionals, allowing timely intervention in areas that lack sufficient psychiatric services. Telepsychiatry can be as effective as in-person consultations, ensuring that high-quality care is extended to remote locations [48].

The integration of machine learning in these digital tools further enhances their capability to predict mental health risks based on the analysis of vast arrays of personal and occupational data. This proactive approach to mental healthcare can potentially prevent the onset of severe mental health conditions. Luxton et al. demonstrated the efficacy of predictive analytics in identifying mental health crises early, which is crucial for timely intervention [49]. For healthcare workers, predictive analytics can identify those at high risk of mental health issues before they manifest, enabling proactive intervention. This approach ensures that resources are allocated efficiently and effectively, focusing on the individuals who are most in need.

In summary, digital health technologies harness the power of data and technology to deliver personalized, accessible, and efficient mental healthcare, particularly to those in remote or underserved areas, underscoring the need for continued innovation and adaptation in healthcare delivery systems.

### 7.4. Patient-Centered Care

Adopting a patient-centered approach in treating healthcare workers, particularly in Latin America, ensures that interventions align with their preferences, needs, and values, thereby improving outcomes and worker satisfaction. This personalized approach acknowledges the unique experiences and stresses of each healthcare worker and tailors interventions to enhance care efficacy and satisfaction.

Studies have highlighted the importance of personalized and patient-centered approaches in diverse healthcare settings, including Latin America. For instance, Carvajal emphasized the need for culturally sensitive and personalized contraceptive counseling for Latina healthcare workers, underscoring the broader application of personalized care principles [50]. Similarly, Espinosa discussed the integration of person-centered and personalized medicine in Latin America, advocating for healthcare that respects the unique contexts and needs of each worker [51].

Such approaches are particularly crucial in regions such as Latin America, where disparities in access to healthcare and cultural diversity present unique challenges. Personalized medicine can significantly affect the effectiveness of mental health interventions among healthcare workers by considering individual and cultural nuances.

## 8. Conclusions

To address the post-COVID-19 mental health challenges of healthcare workers in Latin America, it is crucial to implement a series of strategic interventions that are not only effective but also tailored to meet the unique needs of this diverse group. The critical conditions identified in our review, particularly relevant to the ongoing recovery and adaptation period, underscore the need for enhanced training in resilience and stress reduction. These interventions should be specifically designed to match the psychological profiles and professional stressors of individual workers. Additionally, the provision of basic necessities, such as adequate rest, appropriate accommodations, and essential supplies, must be considered within the context of personalized care, ensuring that these supports are directly aligned with the specific life circumstances and roles of each healthcare professional.

The introduction of specialized training programs must also be adaptive and designed to equip healthcare workers with skills relevant to the evolving demands of their roles during and after the pandemic. Equally important is the role of leadership in fostering a supportive environment through transparent communication and recognition of the efforts and risks undertaken by the healthcare staff.

Acknowledging and addressing moral injury involves more than just recognizing the problem; it requires the implementation of targeted psychological interventions that consider the personal vulnerabilities and professional contexts of the affected individuals. Similarly, the establishment of social support systems and peer interventions should be customized, utilizing insights gathered from ongoing assessments of workers’ mental health states, to provide support that is both relevant and timely.

To normalize and ensure the availability of mental health support programs, it is imperative to integrate adaptive models of care that dynamically respond to the changing needs of healthcare workers. These programs should leverage data-driven insights to continuously refine and personalize support, thereby enhancing the mental health and overall well-being of those on the frontline of healthcare. Through these comprehensive, personalized strategies, we aimed to sustainably support the mental health of healthcare workers in the challenging landscape shaped by COVID-19.

We suggest conducting more longitudinal studies to understand the enduring mental health challenges among healthcare workers in post-COVID-19 Latin America. These studies are essential for revealing the temporal dynamics of mental health, identifying recovery or deterioration periods, and determining when interventions are most effective. They can also assess the long-term impact of interventions and the delayed effects of pandemic stressors, guiding the development of personalized strategies, and informing policies to support healthcare workers’ recovery and adaptation.

To address the mental health challenges faced by healthcare workers in Latin America following the COVID-19 pandemic, it is essential to implement a range of strategic interventions that are effective and tailored to meet the unique needs of this diverse group. The critical conditions identified in our review, particularly relevant to the ongoing recovery and adaptation period, underscore the need for enhanced training in resilience and stress management. These interventions should be designed precisely to match the psychological profiles and professional stressors faced by individual workers. Additionally, the provision of basic needs such as adequate rest, appropriate accommodation, and essential supplies must be considered within the context of personalized care, ensuring that these supports are directly aligned with the specific life circumstances and roles of each healthcare professional.

Specialized training programs must also be adaptable and designed to equip healthcare workers with skills relevant to the evolving demands of their roles during and after the pandemic. Equally important is the role of leadership in fostering a supportive environment through transparent communication and acknowledgment of the efforts and risks undertaken by the healthcare staff.

Addressing moral injury involves more than just recognizing the problem; it requires the implementation of targeted psychological interventions that consider the personal vulnerabilities and professional contexts of the affected individuals. Likewise, the establishment of social support systems and peer interventions should be customized, utilizing insights gathered from ongoing assessments of workers’ mental health states, to provide support that is both relevant and timely.

## Figures and Tables

**Table 1 jpm-14-00680-t001:** Search strategy, terms, and descriptors used.

Database	Search Strategy	Results
PubMed	Title and summary ((Personal OR worker OR professional OR provider) AND (front line, emergency, intensive, internal, first level) AND (mental OR emotional OR psychological OR stress OR exhaustion OR disorder) AND (Latin America OR Hispanics))	7
Scopus	Article Title, Summary, Keywords TITLE-ABS-KEY ((Personal OR healthcare) AND (provider OR staff OR professional OR workers) AND (mental OR emotion OR psychol OR stress OR exhaustion disorders) AND (frontline OR emergency OR intensive OR internal AND (LIMIT-TO (AFFILCOUNTRY, “Latin America”))	8
Google Scholar	Issue: TS = ((provider OR personal OR professional OR workers) AND (front line, emergency, intensive, internal, first level) AND (mental OR emotion OR psychological OR stress OR exhaustion) AND (Latin America, Hispanics)) Refined by: DOCUMENT TYPES: (ARTICLE) AND LAST THREE YEARS OF PUBLICATION Indices: (SCI-EXPANDED, SSCI, A&HCI, CPCI-S, CPCI-SSH, BKCI-S, BKCI-SSH, ESCI, CCR-EXPANDIDO, IC).	12
Total		27

**Table 2 jpm-14-00680-t002:** Inclusion and exclusion criteria.

Variable/Dimension	Inclusion Criteria	Exclusion Criteria
Theme and objective	Psychological and emotional consequences of healthcare assistance after the SARS-CoV-19 pandemic.	Other aspects related to professional activity and health of healthcare personnel (prevention, protection, performance and work organization, work activities and conditions, teleassistance, physical health, epidemiological risks, management and leadership, intervention and support, others)
Design	Quantitative and qualitative	Mixed
Participant population	Healthcare personnel	Non-healthcare personnel, trainees, patients, family members, other profiles
Political, economic, and sociocultural context	Ecuador, Colombia, Perú, Chile, Argentina and Spain	Countries outside the Latin American context
Publication type	Original articles and short reports/articles published in peer-reviewed journals	Letters to the editor, comments, editorials.
Language	Spanish and English	Other language
Publication date	2021, 2022 and 2023	Other dates

**Table 3 jpm-14-00680-t003:** Characteristics of the selected articles.

No.	Authors	Country	Type of Study	Database	Sample	Aim of Study	Mental Disorders
1	Agrest et al., 2022 [14]	Argentina	Exploratory and descriptive	Pubmed	n = 55 (37 female, 16 male)	Characterize the effects of the COVID-19 pandemic on care services for people with psychosocial disabilities	Not mentioned
2	Huamán, 2021 [15]	Peru	Observational, descriptive and transversal	Pubmed	n = 352 (271 female, 81 male)	Determine the relationship between work stress and the mental health of health personnel in the context of the Coronavirus pandemic	Anxiety, depression
3	Gimenez, 2023 [16]	Uruguay	Exploratory and descriptive	Google Scholar	n = 566 (473 female, 89 male	Analyze the experiences of health workers during the pandemic	Anxiety, depression
4	Lemos et al., 2023 [17]	Brazil	Exploratory and descriptive	Pubmed	n = 131	Understand the impacts of the pandemic on mental health and the performance of the psychologist in hospitals, one of the professionals who works in healthcare spaces and who has more closely experienced the suffering of patients and healthcare professionals in the face of COVID-19	Anxiety, insomnia, and stress
5	Moya, 2023 [18]	Peru	Letter to the editor	Google Scholar	Not sample	Discuss and analyze the mental health problems faced by health personnel in Peru during the COVID-19 pandemic	Depression
6	Noguera et al., 2023 [19]	Colombia	Analytical and cross-sectional	Scopus	n = 597 (479 female, 118 male)	Determine the association between sociodemographic, clinical, tobacco and alcohol consumption variables and fear of COVID-19 with the appearance of depression symptoms in a health service provider institution	Depression
7	Oliveira et al., 2023 [20]	Brazil	Cross	Scopus	n = 702	Analyze the prevalence of symptoms of common mental disorders (CMD) in health professionals in Primary Health Care from August to October 2021	Anxiety, depression
8	Naranjo et al., 2021 [21]	Ecuador	Analytical and cross-sectional	Scopus	n = 400 (26% male, 74% female)	Identify the presence of anxious and depressive behaviors in health personnel in Ecuador in the face of the health emergency due to COVID-2019	Anxiety, depression, and post-traumatic stress disorder
9	Santos et al., 2023 [13]	Peru	Observational, descriptive and cross-sectional	Google Scholar	n = 511	Determine the consequences of COVID-19 on the physical, psychological, and social health of obstetricians	Depression, stress, anxiety
10	Serrano et al., 2023 [22]	Spain	Cross-sectional survey	Pubmed	n = 117	Examine the impact of providing medical care during health emergencies caused by viral epidemic outbreaks on the mental health of healthcare workers (HCWs); identify factors associated with a worse impact, and evaluate the available evidence base on interventions to reduce this impact	Depression, anxiety, bipolar disorder

## Data Availability

Not applicable.

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
