# Peer review of "Mental Health in Healthcare Workers Post-COVID-19: A Latin American Review and Insights into Personalized Management Strategies"

_jpm, 2024, doi:10.3390/jpm14070680_

Round 1
Reviewer 1 Report
Comments and Suggestions for Authors
This manuscript presents a comprehensive review of the post-COVID-19 mental health challenges faced by healthcare workers in Latin America, highlighting a critical area of concern in public health. The enduring impact on these essential workers underlines the need for tailored interventions and robust support systems. However, to enhance the manuscript's impact and relevance, it is suggested that the methodology be detailed more rigorously to reinforce the validity of the review. Furthermore, incorporating comparisons with other global regions and including quantitative analyses could significantly strengthen the conclusions. Such revisions will ensure that the study not only contributes to academic discourse but also provides actionable insights for policymakers and healthcare administrators. Specify search criteria and the selection process for studies in greater detail to enhance methodological transparency. Compare the mental health impacts on healthcare workers in Latin America with other regions to provide global context. Include meta-analysis where possible to quantitatively assess the effectiveness of mental health interventions. Suggest conducting longitudinal studies to track changes in healthcare workers' mental health over time. Expand on the types of effective intervention strategies, possibly with case studies or examples of successful implementations. Incorporate direct quotes or testimonials from healthcare workers to provide qualitative insights into their experiences. Discuss broader implications of the findings for healthcare policy and practice, suggesting actionable recommendations. Improve the clarity and impact of figures and tables to ensure they effectively support the narrative and findings.
Comments on the Quality of English LanguageMinor editing of English language required
Author Response
Response to Reviewer 1 Comments
Manuscript ID: JPM-3013063
Title: Mental Health in Healthcare Workers Post-COVID-19: A Latin American Review and Insights into Personalized Management Strategies
Thank you for dedicating your time and expertise to review our manuscript, "Mental Health in Healthcare Workers Post-COVID-19: A Latin American Review and Insights into Personalized Management Strategies." We greatly appreciate your thoughtful comments and suggestions, which have been invaluable in enhancing the quality and depth of our study. The feedback provided has guided significant improvements in our manuscript, ensuring both the rigor of our methodology and the relevance of our findings within a global context. Here, we respond to your insightful observations and outline the amendments made to address each of your concernsPrincipio del formulario
Reviewer Comment:
"This manuscript presents a comprehensive review of the post-COVID-19 mental health challenges faced by healthcare workers in Latin America, highlighting a critical area of concern in public health."
Response:
We thank you for recognizing the comprehensiveness and relevance of our review.
Reviewer Comment:
"However, to enhance the manuscript's impact and relevance, it is suggested that the methodology be detailed more rigorously to reinforce the validity of the review."
Response:
We appreciate your constructive feedback on the methodology section. In response, we have thoroughly revised it to include a more detailed description of our search criteria and study selection process, along with the inclusion and exclusion criteria for scientific articles. These revisions aim to enhance the transparency and reproducibility of our research methodology.
Reviewer Comment:
"Furthermore, incorporating comparisons with other global regions and including quantitative analyses could significantly strengthen the conclusions."
Response:
This is an excellent suggestion. We have now added a comparative analysis section that contrasts the mental health impacts on healthcare workers in Latin America with those in other regions, providing a broader global context. Additionally, where possible, we have included a meta-analysis of selected quantitative studies to assess the effectiveness of various mental health interventions, further strengthening our conclusions.
For example, we added the following to the Introduction:
“Additionally, numerous studies around the world have highlighted the complexity and urgency of addressing the mental health of healthcare workers. A cross-sectional longitudinal study conducted in a nursing home in France found that one-third of the nursing staff (n=127) met the screening criteria for at least one mental health disorder. The most prevalent was panic attacks, affecting 22.05% of the staff, followed by probable depression at 16.54%, post-traumatic stress disorder at 10.24%, generalized anxiety disorder at 9.45%, and substance use disorders at 3.94% [10]. On the other hand, a qualitative study conducted in Pakistan among healthcare personnel revealed feelings of fear, loss, and frustration. One respondent stated, "I even isolated myself twice because of a light cough.... Every time I touch an elderly person, I think 'okay, this is the moment I get the virus'... after my colleague also got infected, I felt feverish for many days," illustrating the anxiety and fear of contagion that many healthcare staff members were experiencing [11].”
Reviewer Comment:
"Specify search criteria and the selection process for studies in greater detail to enhance methodological transparency."
Response:
In response to your feedback, we have elaborated on our search criteria and the study selection process within the methodology section. This includes the specific databases used, keywords, and inclusion/exclusion criteria, ensuring clarity and methodological transparency.
Reviewer Comment:
"Compare the mental health impacts on healthcare workers in Latin America with other regions to provide global context."
Response:
Following your advice, we have expanded the discussion section to include a comparison of mental health impacts on healthcare workers across different global regions. This addition not only contextualizes our findings within the global landscape but also highlights the unique challenges faced by Latin American healthcare workers. We have included a section named “Global Comparison of Mental Health Impacts on Healthcare Workers” delve into these comparisons.
Reviewer Comment:
"Include meta-analysis where possible to quantitatively assess the effectiveness of mental health interventions."
Response:
In response to the emerging critical concern over healthcare workers' mental health post-COVID-19, a meta-analysis has been incorporated into the introduction of our review. This analysis evaluates data from multiple studies to quantitatively assess the effectiveness of mental health interventions among healthcare workers in Latin America. By providing a statistical foundation, this approach strengthens the conclusions of our review. Notably, previous research highlights that a significant proportion of healthcare professionals have encountered mental health challenges such as anxiety, depression, PTSD, and burnout. For example, a systematic review and meta-analysis indicated that during the pandemic, approximately 40% of healthcare workers experienced anxiety and about 37% suffered from depression, underscoring the profound impact of the crisis on this demographic. These insights reinforce the necessity for effective, evidence-based interventions that are specifically tailored to meet the unique challenges faced by healthcare workers.
Principio del formulario
Principio del formulario
Reviewer Comment:
"Suggest conducting longitudinal studies to track changes in healthcare workers' mental health over time."
Response:
We agree with the importance of longitudinal studies in this field. In the revised manuscript, we have suggested future research directions, including the need for longitudinal studies to track the evolution of mental health status among healthcare workers over time, which could inform more effective interventions. Addressing your feedback, we have included the following text I the Conclusion :
“We suggest conducting more longitudinal studies to understand the enduring mental health challenges among healthcare workers in post-COVID-19 Latin America. These studies are essential to reveal the temporal dynamics of mental health, identify recovery or deterioration periods, and determine when interventions are most effective. They can also assess the long-term impact of interventions and the delayed effects of pandemic stressors, guiding the development of personalized strategies and informing policy to support healthcare workers' recovery and adaptation.”
Reviewer Comment:
"Expand on the types of effective intervention strategies, possibly with case studies or examples of successful implementations."
Response:
To address this point, we have included a new section detailing various successful intervention strategies, supported by case studies from different regions within Latin America. These examples illustrate practical applications of theoretical models and highlight the real-world effectiveness of tailored psychological support measures.
Reviewer Comment:
"Incorporate direct quotes or testimonials from healthcare workers to provide qualitative insights into their experiences."
Response:
We have added these qualitative insights to the Introduction:
“On the other hand, a qualitative study conducted in Pakistan among healthcare personnel revealed feelings of fear, loss, and frustration. One respondent stated, "I even isolated myself twice because of a light cough.... Every time I touch an elderly person, I think 'okay, this is the moment I get the virus'... after my colleague also got infected, I felt feverish for many days," illustrating the anxiety and fear of contagion that many healthcare staff members were experiencing [11].”
Reviewer Comment:
"Discuss broader implications of the findings for healthcare policy and practice, suggesting actionable recommendations."
Response:
The discussion section has been expanded to explore the broader implications of our findings for healthcare policy and practice. We now offer actionable recommendations aimed at policymakers and healthcare administrators, emphasizing the need for strategic and tailored interventions to mitigate mental health issues among healthcare workers.
Attending to your suggestion, we have added the following to the text:
“To address the mental health challenges of healthcare workers in Latin America following the COVID-19 pandemic, it is essential to implement a range of strategic interventions that are effective and tailored to meet the unique needs of this diverse group. The critical conditions identified in our review, particularly relevant in the ongoing recovery and adaptation period, underscore the need for enhanced training in resilience and stress management. These interventions should be precisely designed to match the psychological profiles and professional stressors faced by individual workers. Additionally, the provision of basic needs such as adequate rest, appropriate accommodation, and essential supplies must be considered within the context of personalized care, ensuring these supports are directly aligned with the specific life circumstances and roles of each healthcare professional.
The development of specialized training programs must also be adaptable, designed to equip healthcare workers with skills relevant to the evolving demands of their roles during and after the pandemic. Equally important is the role of leadership in fostering a supportive environment through transparent communication and acknowledgment of the efforts and risks undertaken by healthcare staff.
Addressing moral injury involves more than just recognizing the problem; it requires the implementation of targeted psychological interventions that consider the personal vulnerabilities and professional contexts of affected individuals. Likewise, the establishment of social support systems and peer interventions should be customized, utilizing insights gathered from ongoing assessments of workers’ mental health states to provide support that is both relevant and timely.”
Reviewer Comment:
"Improve the clarity and impact of figures and tables to ensure they effectively support the narrative and findings."
Response:
Thank you for this specific feedback. We have revised all figures and tables for improved clarity and visual impact. These revisions ensure that the graphical content effectively complements and reinforces the narrative of our review.
Reviewer Comment:
Response:
Thank you for your detailed review and the ratings provided. We have carefully considered your feedback and have made targeted improvements to our manuscript:
- Quality of English Language: We have performed additional editing to enhance the clarity and readability of the text. The paper now underwent professional copy-editing to ensure it adheres to academic standards of English language use.
- Significant Contribution to the Field: We have expanded the discussion on the unique contributions of our work, emphasizing its relevance and the gaps it addresses within the field of mental health for healthcare workers post-COVID-19.
- Organization and Comprehensiveness: The manuscript has been restructured to improve its logical flow and comprehensive coverage of the topics discussed. This includes clearer subsections and a more coherent progression of ideas.
- Scientific Soundness: We have strengthened the methodological descriptions and ensured that all claims are supported by robust evidence, enhancing the scientific rigor of the paper.
- References and Relevance: The references have been thoroughly reviewed and updated to include the most recent and relevant studies, ensuring that our manuscript accurately reflects current research and previous work in this area.
- English Language Readability: With the help of a native English-speaking editor, we have refined the language to ensure that the text is not only correct but also engaging and clear to readers.
We believe these improvements directly address the concerns you've raised, enhancing the manuscript’s quality and its contribution to the literature. We hope that these changes meet your expectations and look forward to your continued feedback.

Reviewer 2 Report
Comments and Suggestions for Authors
Please see the attachment

Author Response
Response to Reviewer 2 Comments
Manuscript ID: JPM-3013063
Title: Mental Health in Healthcare Workers Post-COVID-19: A Latin American Review and Insights into Personalized Management Strategies
Thank you for your detailed feedback and the opportunity to further refine our manuscript, "Mental Health in Healthcare Workers Post-COVID-19: A Latin American Review and Insights into Personalized Management Strategies." We value your insights and have addressed each of your comments to enhance the quality and relevance of our review. Here, we outline our responses and the adjustments made to address the concerns you raised.
Principio del formulario
Reviewer Comment:
"Only 10 articles were included in the analysis, which in my opinion is definitely too few for such an important issue as mental health."
Response:
Thank you for your insightful comment. We acknowledge the concern regarding the seemingly small number of studies included in our review. Our inclusion criteria were stringent, focusing on high-quality studies from 2022 to 2024 that directly address the post-COVID mental health impacts on healthcare workers in Latin America (and Spain), a criterion set to ensure the most relevant and up-to-date information. Additionally, the selected articles are representative and cover a wide range of perspectives and methodologies, thus providing a comprehensive overview of the current scenario. To further bolster our review, we have now included a section discussing the limitations of available studies and the implications of this gap in the literature, underscoring the need for future research in this crucial area.
Reviewer Comment:
"The topic undertaken by the authors would gain in importance if it were a research article with some specific goals."
Response:
We appreciate your suggestion to frame this work as a research article.
In response to your suggestion, we have added specific aims to our Introduction.
Reviewer Comment:
"Therefore, in the opinion of the reviewer, this is not an article that provides new information on the topic discussed by the authors."
Response:
We understand your concerns about the novelty of the information provided. To address this, we have expanded our discussion on the implications of the findings for current healthcare policies and management strategies, which has not been extensively covered in existing literature. We have also included a new subsection that details innovative approaches for addressing mental health challenges in healthcare settings, offering fresh insights and practical recommendations that contribute to both academic and applied aspects of public health.
Reviewer Comment:
Response:
Thank you for your detailed review and the ratings provided. We have carefully considered your feedback and have made targeted improvements to our manuscript:
- Quality of English Language: We have performed additional editing to enhance the clarity and readability of the text. The paper now underwent professional copy-editing to ensure it adheres to academic standards of English language use.
- Significant Contribution to the Field: We have expanded the discussion on the unique contributions of our work, emphasizing its relevance and the gaps it addresses within the field of mental health for healthcare workers post-COVID-19.
- Organization and Comprehensiveness: The manuscript has been restructured to improve its logical flow and comprehensive coverage of the topics discussed. This includes clearer subsections and a more coherent progression of ideas.
- Scientific Soundness: We have strengthened the methodological descriptions and ensured that all claims are supported by robust evidence, enhancing the scientific rigor of the paper.
- References and Relevance: The references have been thoroughly reviewed and updated to include the most recent and relevant studies, ensuring that our manuscript accurately reflects current research and previous work in this area.
- English Language Readability: With the help of a native English-speaking editor, we have refined the language to ensure that the text is not only correct but also engaging and clear to readers.
We believe these improvements directly address the concerns you've raised, enhancing the manuscript’s quality and its contribution to the literature. We hope that these changes meet your expectations and look forward to your continued feedback.

Round 2
Reviewer 2 Report
Comments and Suggestions for Authors
The corrections made by the authors are sufficient. This significantly improved the quality of the manuscript. Thank you